# Analysis of Chemical Composition of Extractives by Acetone and the Chromatic Aberration of Teak (*Tectona Grandis* L.F.) from China

**DOI:** 10.3390/molecules24101989

**Published:** 2019-05-23

**Authors:** Hongyun Qiu, Ru Liu, Ling Long

**Affiliations:** 1Research Institute of Forestry New Technology, Chinese Academy of Forestry, Haidian 100091, Beijing, China; qiuqiuiuu@163.com; 2Research Institute of Wood Industry, Chinese Academy of Forestry, Haidian 100091, Beijing, China

**Keywords:** acetone extractive, chemical composition, teak, chromatic aberration

## Abstract

In order to clarify the chemical color change of teak (*Tectona grandis* L.F.), the difference of chemical composition between the heartwood and sapwood of teak was investigated by gas chromatography–mass spectrometry (GC-MS) based on the acetone extractive compounds. The results showed that the difference in content of the main components between heartwood and sapwood was not obvious. However, the amount of extractives in heartwood was higher than that in sapwood, especially for phenols, quinones, and ketones. The most obvious different substances in the acetone extractive between heartwood and sapwood were 4-tert-butyl-2-phenyl-phenol,2-methyl-anthraquinone, and 2,3-dimethyl-1,4,4a,9a-tetrahydro-9,10-anthracenedione, which might be the main composition for the chromatic aberration of teak. This paper focuses on a preliminary study and further work such as high-performance liquid chromatography (HPLC) with ultraviolet photometric detector (UV)/mass spectrometry (MS) will be carried out.

## 1. Introduction

Teak (*Tectona grandis* L.F.), as one of the most famous timber species all over the world, is distributed from 73° E in India to 104°30′ E in Thailand, and from the island of Java at 8°45′ S to the low hills and plains which is below 800 m in northern Myanmar in terms of its origin [1]. It is widely used in boat masts and decks, furniture, railway sleepers and other applications due to its important advantages such as high hardness and durability, fine texture, beautiful texture, antiseismic performance, small shrinkage, etc. [2]. However, poor visual appearance in green teak is common by reason of the uneven and unstable surface color [3]. To provide basic information for the wood industry, the physical and mechanical properties and compositions of teak have been analyzed by many researchers [4,5,6,7]. A lot of research has been done into the breeding, plantations and growth of teak so as to develop high productivity and quality varieties [8]. Abundant genetic variation in photosynthetic traits and growth among the teak clones, relations between trace element in cultivated soil or bent sections and properties were obtained [9,10], for instance. However, the chemical compositions were rarely reported. As the chemical composition of wood is one of the most important factors for its properties, it should not be ignored.

Extractives were used to identify different substance in wood [11], drug [12], food [13] and the like, although they account for small parts of the wood, and gives the wood its unique properties [14,15,16]. The reason why teak is loved by consumers is inseparable from its attractive appearance, and the unique color, which is highly correlative with the extractives. At present, the research on the chemical composition of teak is mostly around lignin, cellulose, and hemicelluloses [17]. Few studies are related to wood extractives, let alone to the extractives of teak in China [3,18,19,20,21]. Therefore, the analysis of the chemical composition of teak, especially its extractives, can provide a theoretical basis for the rules of the color change and color matching technology of teak, which has great research value in its practical application. The chemical constituents of teak from Yunnan province in China were researched quantitatively in this study. Acetone was used to extract the heartwood and sapwood of teak. The extractives were respectively analyzed by gas chromatography–mass spectrometry (GC-MS) to find the difference in composition between the two parts.

## 2. Materials and Methods

### 2.1. Materials

Eighteen-year-old teak was taken from Yingjiang, Dehong, Yunnan province, China. A disc with a diameter of about 20 cm in 5 cm thickness was taken from a 100 cm-long teak log. Six logs were chosen in total, and the dics were selected from different position of the logs, namely, the top, middle, and bottom. They were exposed naturally for air-dried. The experiment reagents used in this study are listed in Table 1.

### 2.2. Determination of Chemical Composition

The relative Chinese standards were found based on the book of Shi and HE [22]; 5 × 5 × 50 mm^3^ sections of wood were chopped along the vertical direction of 6 teak discs. The sapwood and heartwood were ground by a pulverizer (1HP desktop high-speed pulverizer RT-34 from Rong Tsong Precision Technology Co., Taiwan, China) into mixture powder from 6 discs with 40–60 mesh respectively. The moisture of sapwood and heartwood of teak were determined by reference of GB/T 2677.2-1993 “Fibrous raw material–Determination of moisture content”.

Determination of α-cellulose and hemicellulose content were measured by reference of GB/T 744-1989 “Pulps–Determination of α-cellulose” and GB/T 2677.10-1995 “Fibrous raw material– Determination of holocellulose”, respectively.

The determination of lignin was carried out in accordance with GB/T 2677.8-1994 “Fibrous raw material–Determination of acid-insoluble lignin”.

The pH value was measured referring to GB/T 6043-1999 “Method for determination of pH of wood”. 3 g wood flour sample was extracted by alcohol-benzene, 1% NaOH, cold water, hot water, and the acetone, respectively referring to GB/T 10741-1989 “Pulps–Determination of alcohol-benzene soluble”, GB/T 2677.5-1993 “Fibrous raw material Determination of 1% sodium hydroxide solubility”, GB/T 2677.4-1993 “Fibrous raw material–Determination of water solubility” and GB/T 2677.6-1994 “Fibrous raw material–Determination of solvent extractives”. After the extraction complete, the extractives were transferred to 50 ml volumetric flasks and bring to volume.

### 2.3. Total Phenol Content

The total phenol content was measured with the cold water extractives and hot water extractives by spectrophotometry. The standard solution of gallic acid was detected by wavelengths at 780 nm. Regression analysis of the data to obtain the regression equation is: *y* = 0.00059 + 0.1149*x* (*y* is absorbance, *x* is the total phenol content in the solution).

### 2.4. GC-MS Analysis

The analysis was carried out by Fisher TSQ Quantum GC (gas chromatography) (Thermo Fisher, Massachusetts, USA). The chromatographic conditions were Agilent (Agilent Technologies Inc., California, USA) DB-1MS (60 m × 0.25 mm × 0.25 μm) with column flow rate of 1.0 ml/min. The inlet temperature was 250 °C, and split ratio was set at 15:1. The initial temperature was 40 °C and rose to 280 °C at 10 °C/min, then retained for 15 min. The mass spectrometry conditions were an EI (Electronic ionization) ionization source with electron bombardment energy of 70 eV, ion source temperature of 250 °C, filament current of 100 μA, and transmission line temperature of 250 °C. The compounds were identified by comparing the MS (mass spectrometry) spectra to the National Institute of Standards and Technology (NIST) library (http://webbook.nist.gov/chemistry/, Maryland, USA) with retention time and indices.

## 3. Results and Discussion

### 3.1. Components Analysis in Sapwood and Heartwood of Teak

Lignin and extractives in wood are the main reasons for the different colors of wood. The main component contents and pH value of heartwood and sapwood of teak are shown in Table 2.

It can be found that the main component contents of teak between heartwood and sapwood were not significant. Both of them were in the weak acidic range. The moisture of heartwood was slightly lower than that of the sapwood. In addition to the structural difference of lignin, the extractives had a certain effect on the color of the teak wood. To obtain natural extracts, methods using solvents are the most common, in particular for herbs solid–liquid extraction, with the simplest being maceration [23]. As we know, the extractives extracted by different solvents from wood are different in type and content [24]. The water-soluble substances in wood are tannins, pigments, carbohydrates, plant alkaloid, cyclitol and some inorganic salts, while hot 1% sodium hydroxide solution can dissolve not only the water extractives, part of lignin, pentosan, hexosan and resin acids in wood, but also the furfural acid and the degraded components caused by light, heat, oxidation and bacteria. The alcohol–benzene solution not only has the ability of benzene to dissolve fats and waxes, but also has an excellent solubility to dissolve water in any proportion. So it can dissolve resins, fats and waxes, and also tannins, pigments, etc. Acetone extractives are mainly composed of fats, waxes, fatty acids, terpenoids, phenolic compounds, etc. [25,26,27,28]. The contents of the following extractives in sapwood and heartwood of teak are shown in Table 3.

As expected, the content of extractives (Table 3) in hot water was more than that in cold water, and the same phenomenon occurred in the total phenol. The total phenol, which exhibited good biological activities, was detected more in heartwood than sapwood. Obviously, the content in alcohol–benzene and acetone extractives between heartwood and sapwood was quite different. The chemical compositions extracted from alcohol–benzene were higher than others. Deng [29] obtained that the acetone can serve as an alternative for alcohol–benzene. Therefore, the acetone extractives in heartwood and sapwood of teak were analyzed by the gas chromatography–mass spectrometry (GC–MS), which can better identify the chemicals. By comparing the distinctions between the two parts, the reason why the color between heartwood and sapwood of teak could be understood.

### 3.2. GC–MS Analysis of Acetone Extractives

As the typical substance of teak, 2-methyl-anthraquinone was suspected to be the reason for the special color and surface properties of teak. According to the external standard method, the area of the standard solution at different concentration was detected by GC–MS. The standard curve drawn according to the test results is shown in Figure 1. Regression analysis of the data to obtain the regression curve of the standard curve was: *y* = 1.1991 × 107 × *x* (*y* is the peak area, *x* is the content of 2-methyl-anthraquinone) with R2 of 0.9987. It can be seen from Figure 1 that the standard curve was approximately a straight line, which meant that the amount of the standard solution had a good linear relationship with the peak area.

The quantitative analysis of the other substance in the acetone extractive detected by GC-MS depended on the relationship of the peak areas between them and the 2-methyl-anthraquinone. The results of GC-MS analysis of acetone extracts in heartwood and sapwood of teak are shown in Figure 2 and Table 4, and the most prominent compositions are marked in Figure 2 in addition.

The acetone extractives in heartwood were higher in composition and content than those in sapwood (Figure 2). A total of 49 components (Table 4) were detected in the acetone extractives of the heartwood, including 10 kinds of alkanes, 2 kinds of olefins, 1 benzene series, 1 alcohol, 4 kinds of aldehydes, 8 kinds of ketones, 2 kinds of acids, 3 kinds of easters,1 anhydride,3 kinds of phenols, 7 kinds of hydrazines and 7 kinds of heterocyclics; while 26 kinds of components were detected in the acetone extractives of the sapwood, including 7 kinds of alkanes, 2 kinds of olefins, 1 alcohol, 1 aldehyde, 4 kinds of ketones, 1 acid, 1 ester, 2 kinds of phenols, 5 kinds of hydrazines and 2 kinds of heterocyclics. The benzene series and acid anhydrides were not detected in the sapwood. Except for acetonyldimethylcarbinol and estriol, the rest components in sapwood were less than that in the heartwood. Although some extractives in the heartwood were undetected, most of the constituents with higher content can be detected in the sapwood.

The components in acetone extractives of heartwood and sapwood are accumulated according to the type and shown in Table 5. The percentages of the various contents in heartwood and sapwood are shown in Figure 3.

Alkanes, ketones, phenols and quinones (Table 5 and Figure 3) in acetone extractives had relatively high contents in heartwood, while in sapwood, the ketones, phenols and quinones were much less. The content of benzene, alcohols, acids, esters, acid anhydrides and heterocyclics were very low both in the heartwood and sapwood, while the substances of olefins, aldehydes and phenols accounted for a certain proportion in the heartwood but were hardly present in the sapwood. In particular, the phenols, which had the highest content in the heartwood, was rarely found in the sapwood. Among the above content, the most prominent chemical compositions are shown in Table 6.

Components like lappaol, deoxylactam and its isomers, such as squalene, chloranol, palmitic acid, etc. were all detected in teak indigenous to Java Island, Yogyakarta and Panama. Similar substances were found in this research. Due to the different origins, the content of each component was different. The tectol with higher content detected in teak from Indonesia, did not appear in this experimental result; 4-tert-butyl-2-phenylphenol and 2,3-dimethyl-1,4,4a,9a-tetrahydro-9,10-nonanedione, which had a high content in this research, were undetected in teak from Indonesia and Panama. Although ketones appeared less than in other literature, 2,3-dimethyl-1,4,4a,9a-tetrahydro-9,10-nonanedione was similar to lappaol, deoxylapram and its isomers in structure. Extractives of the teak grown in the B region of Panama, increased in the amount from the sapwood to heartwood, which was consistent with the results obtained in this paper that the heartwood extractives were greater than the sapwood [30,31]. It should be noted that some compounds might not belong to wood extractives, such as estriol, a mammalian hormone, and antioxidant of butyl(2-chlorocyclohexyl) methyl phthalate. These compounds might be the dissolved impurity from the plastic lips.

### 3.3. Discussion in the Chromatic Aberration Between Heartwood and Sapwood

Although the proportion of extractives in wood is small, they play an important role in the particular characteristics such as the color and smell of the wood, which is highly related to wood discoloration [32,33]. The extractives contained chromatic substances such as pigments, tannins and resins. Most of these color-related components had phenolic hydroxyl groups, carbonyl groups, double bond structures, etc. [34]. In the acetone extractives of heartwood (Figure 3), the substances with these structures, such as olefins and phenols, were more than those in the sapwood, which may be one of the reasons why the color of heartwood was darker than the sapwood. From the differences in compositions, it can be assumed that phenols, quinones, and ketones in teak may be the main extractives that caused heartwood to be darker than sapwood. Also, the substances with significant differences in the content of heartwood and sapwood (Table 6) have the chemical groups related to wood color such as phenolic hydroxyl group, carbonyl group and double bond, which might be the main composition for the chromatic aberration of teak. 

The increase of chromophoric group and auxochrome group in the extractives after illumination caused a change in wood color [3]. The yellowness of the surface may be related to the amount of the 2-methylindole [35]. But due to the small amounts of extractives, the different extractive effects under different solvents which may change the compositions in the same wood and the relationship between tree-age and extractives, the relationship between extractives and discoloration, and the mechanism of discoloration of teak, have not been well explained. Therefore, further study will focus on the mechanism of discoloration of teak.

## 4. Conclusions

The difference in content of the main components between heartwood and sapwood was slight. They both were appeared to be weakly acidic. The moisture of heartwood was slightly lower than that of sapwood. The content in alcohol–benzene and acetone were obviously different between heartwood and sapwood.

The acetone extractives in heartwood were higher in composition and content than those in sapwood; 49 components were detected in the acetone extractive of the heartwood, and 26 components were detected in the sapwood. The high content in the teak heartwood was mostly detected in the sapwood.

In the acetone extractives, the substances of phenols, quinones and ketones in sapwood were obviously less than those in the heartwood. These substances may be the main components that made the heartwood darker than the sapwood. 

The most obvious distinct substances in acetone extractives between heartwood and sapwood were 4-tert-butyl-2-phenyl-phenol, 2-methyl-anthraquinone and 2,3-dimethyl-1,4,4a, 9a-tetrahydro-9, 10-anthracenedione#.

This paper on the issue was a preliminary study. Since GC–MS was not the most accurate analysis method, a liquid chromatography system like high-performance liquid chromatography (HPLC) with UV–MS will be used to confirm the possible substance for the chromatic aberration in teak from China in a further study.

## Figures and Tables

**Figure 1 molecules-24-01989-f001:**
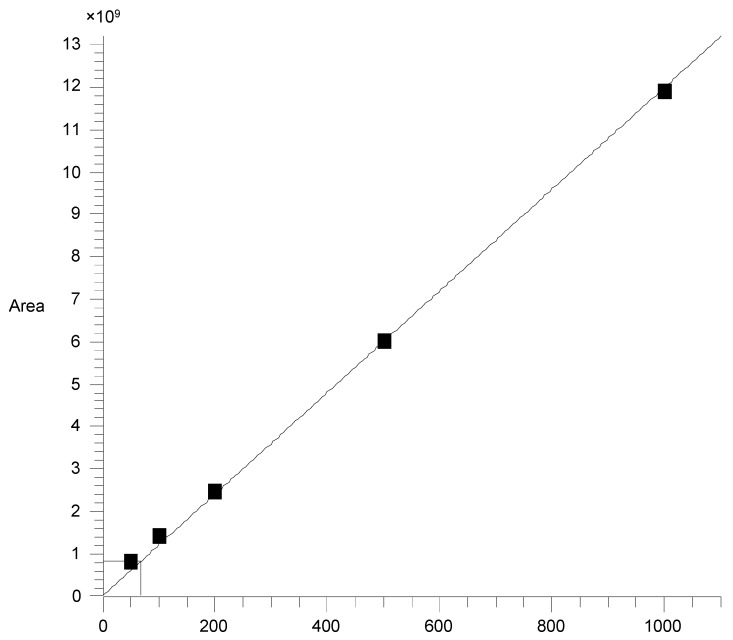
Standard curve about 2-methyl-anthraquinone and peak area.

**Figure 2 molecules-24-01989-f002:**
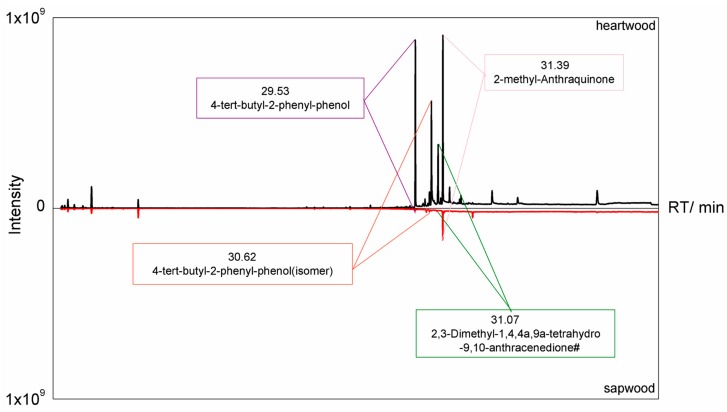
Gas chromatography–mass spectrometry (GC–MS) chromatogram of acetone extractive in the heartwood and sapwood of teak.

**Figure 3 molecules-24-01989-f003:**
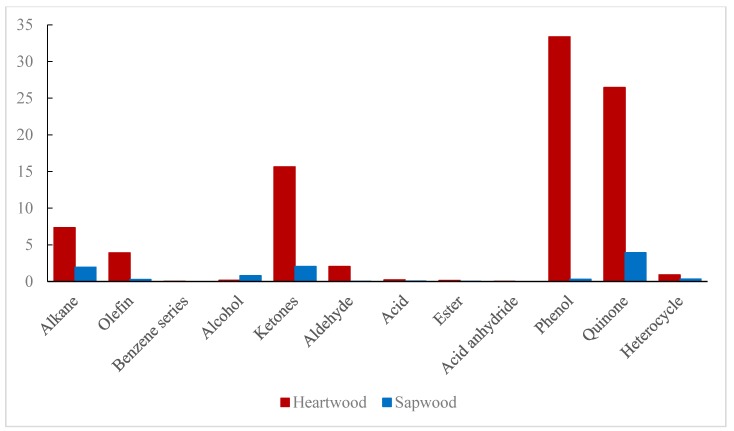
Contents of various compounds in acetone extractives in the heartwood and sapwood of teak.

**Table 1 molecules-24-01989-t001:** Experiment reagent.

Experiment Reagent	Manufacturers
Distilled water	Guangzhou Watson’s Food and Beverage Co., Ltd., Guangzhou, China
Sodium hydroxide (analytical grade)	Beijing Chemical Plant, Beijing, China
Sulfuric acid (analytical grade)	Beijing Chemical Plant, Beijing, China
Benzene	Tianjin Fuchen Chemical Reagent Factory, Tianjin, China
Methanol (HPLC(high-performance liquid chromatography) grade)	Mreda Technology Inc, California, USA
Acetone (HPLC grade)	Duksan Pure Chemicals Co. Ltd., Ansansi Gyunggido, Korea

**Table 2 molecules-24-01989-t002:** Chemical components of woods.

	Moisture (%)	Acid-Insoluble Lignin (%)	Holocellulose (%)	α-Cellulose (%)	pH Value
Heartwood	3.90	26.76	58.27	34.97	5.79
Sapwood	4.25	27.73	62.23	35.53	5.84

**Table 3 molecules-24-01989-t003:** Contents of partial extractives in heartwood and sapwood of teak.

	Hot Water (%)	Cold Water (%)	1% NaOH (%)	Alcohol-benzene (%)	Acetone (%)	Total Phenol (%)	
						Hot Water	Cold Water
Heartwood	10.22	6.40	22.34	14.54	10.13	1.35	0.71
Sapwood	9.36	6.07	24.06	4.67	2.37	0.93	0.33

**Table 4 molecules-24-01989-t004:** GC–MS analytical results of acetone extractive compounds in heartwood and sapwood of teak.

NO.	Categories	Retention Time (min)	Retention Index	Component	GC Content (μg/g)	
					Heartwood	Sapwood
1	Alkane	5.61	570	2-methyl-pentane	45.01	13.89
2		5.79	584	3-methyl-pentane	56.94	17.89
3		6	601	N-hexane	207.26	73.03
4		6.44	630	Methyl-cyclopentane	88.56	27.99
5		7.02	664	Cyclohexane	65.32	19.78
6		7.61	691	Isooctane	589.54	129.64
7		8.65	736	2,2,3-trimethylpentane	7.54	/
8		9.06	630	2,3,4-trimethylpentane	11.72	4.00
9		28.38	1360	5,8-diethyldodecane	4.82	/
10		28.53	1387	9-n-hexylheptadecane	5.02	/
11	Olefin	8.99	750	2,3-dimethyl-1-hexene	5.65	4.21
12		41.85	2835	All-trans-squalene	569.65	36.83
13	Benzene series	28.16	1323	1,3,5-triisopropyl-benzene	4.61	/
14	Alcohol	33.41	2970	Estriol	23.66	118.49
15	Ketones	10.77	811	Acetonyldimethylcarbinol	163.09	182.46
16		29.7	1927	6-(1-Hydroxymethylvinyl)-4,8a-dimethyl-3,5,6,7,8,8a-hexahydro-1H-naohthalen-2-one	13.40	2.53
17		30.19	1941	2-Acetyl-3-methyl-3-phenyl-2,3-dihudro-5H-indazol-5-one	145.50	/
18		31.07	1966	2,3-Dimethyl-1,4,4a,9a-tetrahydro-9,10-anthracenedione	1228.70	30.31
19		31.25	1971	Tetracyclo [10.2.1.0(2,11).0(4,9)] petadeca-2(11),6,13-triene-5,8-dione	107.40	/
20		31.85	1988	2-(3-Hydroxyphenyl)-1H-indene-1,3(2H)-dione	320.73	4.42
21		32.14	1966	2-Acetyl-3-methyl-3-phenyl-2,3-dihydro-5H-indazol-5-one	58.41	/
22		36.46	3387	2-(2-Nitro-1-phenyl-2-propenyl)cyclohexanone	266.51	/
23	Aldehyde	22.17	1285	1,3-benzodioxole-5-carboxaldehyde (Piperonal)	20.94	/
24		26.49	1640	(2E)-3-(1,3-benzodioxol-5-yl)-2-propenal	47.10	/
25		32.62	2679	9,10-dioxo-9,10-dihydro-1-anthracenecarbaldehyde	170.20	4.21
26		33.15	2846	1-methyl-1,2,3,4,4a,9,10,10a-octahydro-1-phenanthrenecarbaldehyde	62.81	/
27	Acid	7.2	702	Propanoic acid	9.63	/
28		29.17	1912	1,4-Dihydroxy-3-(3-methyl-2-butenyl)-2-naphthoic acid	23.66	7.16
29	Ester	8.33	726	Methyl isocyanate	11.72	/
30		12.7	878	Acrylic acid butyl ester	3.35	/
31		28.77	1901	Butyl(2-chlorocyclohexyl) methyl phthalate	4.82	2.95
32	acid anhydride	21.61	1322	1,3-isobenzofurandione (Phthalic anhydride)	3.35	/
33	Phenol	27.31	1675	4-((1E)-3-hydroxy-1-propenyl)-2-methoxyphenol	7.33	/
34		29.53	1922	4-tert-butyl-2-phenyl-phenol	2675.75	23.99
35		30.62	1953	4-tert-butyl-2-phenyl-phenol (isomer)	2230.46	21.68
36	Quinone	23.31	1427	1,4-naphthoquinone	20.73	/
37		24.84	1469	Menadione	15.28	/
38		30.05	1937	Anthraquinone	101.54	2.10
39		30.48	1949	2-hydroxy-3-(3-methyl-2-butenyl)-1,4-Naphthoquinone (Lapachol)	227.36	31.36
40		31.39	1975	2-methyl-anthraquinone	3019.30	516.46
41		32.53	2855	1-Hydroxy-4-methylanthra-9,10-quinone	94.21	14.73
42		34.73	3064	2-(Hydroxymethyl)anthraquinone	419.96	13.05
43	Heterocycle	26.09	1603	3,4-methylenedioxybenzhydrazide	3.56	/
44		26.27	1612	Dibenzo-p-dioxin	4.61	/
45		28.3	1844	5-methoxy-7-phenyl-bicyclo [3.2.0] hept-2-en-6-one, (Z, exo+ endo)	14.86	/
46		28.61	1876	1,2-benzisothiazol-3-amine tbdms	6.70	/
47		28.92	1905	2-Hydroxydibenzofuran	26.59	4.00
48		30.28	1944	4a-methyl-1-methylene-1,2,3,4,4a,9,10,10a-octahydrophenanthrene	50.87	45.25
49		35.79	3241	Lochnerine	26.80	/

**Table 5 molecules-24-01989-t005:** The content of acetone extractive in heartwood and sapwood of teak (by category).

	Heartwood (μg/g)	Sapwood (μg/g)
Alkane	1081.73	286.22
Olefin	575.30	41.04
Benzene series	4.61	0.00
Alcohol	23.66	118.49
Ketones	2303.73	301.79
Aldehyde	301.05	4.21
Acid	33.29	7.16
Ester	19.89	2.95
Acid anhydride	3.35	0.00
Phenol	4913.54	45.67
Quinone	3898.38	577.70
Heterocycle	133.99	49.25

**Table 6 molecules-24-01989-t006:** The compounds in the heartwood and sapwood of teak with obvious difference in content.

	Structures	Heartwood ug/g	Sapwood ug/g
4-tert-butyl-2-phenyl-phenol	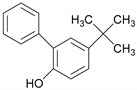	4906.21	45.67
2-methyl-Anthraquinone	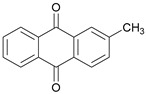	3019.30	516.46
2,3-Dimethyl-1,4,4a,9a-tetrahydro-9,10-anthracenedione#	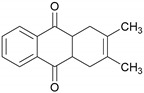	1228.70	30.31

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
