# Peer review of "Analysis of Chemical Composition of Extractives by Acetone and the Chromatic Aberration of Teak (Tectona Grandis L.F.) from China"

_molecules, 2019, doi:10.3390/molecules24101989_

Round 1

Reviewer 1 Report

While the aim of this study has its merit, it has several fundamental shortcomings that prohibit to draw any sound conclusions from it.

The project is based on a single wood disk - for a reliable analysis, a larger populations must be employed.

Several extractions with different solvent are performed, which is definitely necessary in extractive analysis, but then only a single one is analysed. In extractive analysis, a single extract will never give the full picture, and a lot of information is lost by not analysing the other extracts. The stated reason to ignore the other extracts is "the component was difficult to separate".

GC conditions are inadequte for extractives analysis: an inlet temperature of 250°C and a maximum oven temperature of 280°C will not elute a large portion of the analytes. Extractive analysis is usually performed at high-temperature systems allowing temperatures up to 400°C. No derivatization of the sample is mentioned, which again means that only very apolar compounds will be detected. No mention is made how compounds were identified. A simple database search is insufficient, at least retention indices must be taken into account as well

As a side note, in a project that aims at determining chromophoric compounds, GC-MS might not be the best choice. A liquid chromatography system (LC, SFC or HPTLC) with UV/Vis and MS would identify the chromophoric compounds and their structure withoug discriminating by boiling point and polarity.

A large portion of the detected compounds appear to be misattributed (estriol, a mammalian pregnancy hormone) or artifacts from sample preparation and the employed chemicals (the classics: BHT, phthalates, probably all of the alkanes). Unfortunately, the major peak of 4-tert-butyl-2-phenyl-phenol is among them. Reporting the results of extractives must really take into account that several compounds are involuntarily added by solvents and reagents, and these artifacts must be detected and labelled. A blank run and thotough check of the detected compounds would help in doing so.

The anthraquinones might indeed contribute to the colour, but they are very prone to oxidation and isomerization due to changes in pH. To obtain a really reliable result of their condition in vivo, sample extraction would have to be very careful.

I could not understand how the authors obtained quantitaive values from calibrating a single standard. Response factors vary a lot in GC-MS, and can not reliably be predicted.

There are a several typos in chemical names, and it is not clear to me how the components listed in lines 161 and 162 are isomers (chloranol (hypochloric acid), hexadecanoic acid and deoxylactam? Which deoxylactam of the many?).

Author Response

We must thank you for the valuable comments and suggestions, which helped improve our manuscript greatly. Please do forward our heartfelt thanks to the reviewers. Based on the comments we received, careful modifications have been made to the manuscript. All changes were marked in text. We hope that the revised manuscript answered the questions. Below you will find our point-by-point responses to the comments/ questions:

To Reviewer 1:

While the aim of this study has its merit, it has several fundamental shortcomings that prohibit to draw any sound conclusions from it.

The project is based on a single wood disk - for a reliable analysis, a larger populations must be employed.

Thanks. We are sorry for making a mistake of using a single wood disc. The sample was prepared from powder of 6 discs. We added the description in line 54 that “A disc with a diameter of about 20 cm in 5 cm thickness was taken from the 100 cm long teak log. 6 logs were chosen in total, and the discs were selected from different position of the logs, namely, the top, middle, and bottom. They were exposed naturally for air-dried.” Please check.

Several extractions with different solvent are performed, which is definitely necessary in extractive analysis, but then only a single one is analysed. In extractive analysis, a single extract will never give the full picture, and a lot of information is lost by not analysing the other extracts. The stated reason to ignore the other extracts is "the component was difficult to separate".

Thanks. We have extracted the wood with hot water, cold water, 1% NaOH, alcohol-benzene, and acetone. As we mentioned in line 119, the content of extractives in hot water, cold water, and 1% NaOH were almost the same between sapwood and heartwood. Difference was found in the alcohol-benzene and acetone groups. We are sorry for the wrong reason of only analyzing acetone extractives. We have corrected it in line 125. “Deng [29] obtained that the acetone can serve as an alternative for alcohol-benzene.” Please check.

Ref:

29. Deng, Z. Development of an alternative solvent to replace benzene in the determination of organic soluble extractives in wood. World Pulp Paper 2006, 25, 32-34.

GC conditions are inadequte for extractives analysis: an inlet temperature of 250°C and a maximum oven temperature of 280°C will not elute a large portion of the analytes. Extractive analysis is usually performed at high-temperature systems allowing temperatures up to 400°C. No derivatization of the sample is mentioned, which again means that only very apolar compounds will be detected. No mention is made how compounds were identified. A simple database search is insufficient, at least retention indices must be taken into account as well

Thanks. The GC condition was set according to other researchers such as Yin et al. [11], where less thermal decomposition occurred. As for your suggestion, we added the retention indices in Table 4. Please check.

Ref:

11. Yin, X.; Huang, A.; Zhang, S.; Liu, R.; Ma, F. Identification of Three Dalbergia Species Based on Differences in Extractive Components. Molecules 2018, 23, 2163.

As a side note, in a project that aims at determining chromophoric compounds, GC-MS might not be the best choice. A liquid chromatography system (LC, SFC or HPTLC) with UV/Vis and MS would identify the chromophoric compounds and their structure withoug discriminating by boiling point and polarity.

Thanks for your sincere suggestion. Due to the limitation of correction time of 10 days, we will take the HPLC with UV/MS for the chromatic compounds in further study.

A large portion of the detected compounds appear to be misattributed (estriol, a mammalian pregnancy hormone) or artifacts from sample preparation and the employed chemicals (the classics: BHT, phthalates, probably all of the alkanes). Unfortunately, the major peak of 4-tert-butyl-2-phenyl-phenol is among them. Reporting the results of extractives must really take into account that several compounds are involuntarily added by solvents and reagents, and these artifacts must be detected and labelled. A blank run and thotough check of the detected compounds would help in doing so.

Thanks. These compounds might not belong to the wood extractives. We have mentioned that in line 357. “It should be noted that some compounds might not belong to wood extractives, such as estriol, a mammalian hormone, and antioxidant of butyl(2-chlorocyclohexyl) methyl phthalate. These compounds might be the dissolved impurity from the plastic lips.” Please check.

The anthraquinones might indeed contribute to the colour, but they are very prone to oxidation and isomerization due to changes in pH. To obtain a really reliable result of their condition in vivo, sample extraction would have to be very careful.

Thanks. The anthraquinones extracted from wood were stored in the acetone liquid, which was sealed up to avoid oxidation before analyzing carefully.

I could not understand how the authors obtained quantitaive values from calibrating a single standard. Response factors vary a lot in GC-MS, and can not reliably be predicted.

Thanks. We added details for the obtaining of quantitative values in line 73 that “3 g wood flour sample was extracted by alcohol-benzene, 1% NaOH, cold water, hot water, and the acetone, respectively…” and line 77 “After the extraction complete, the extractives were transferred to 50 ml volumetric flasks and bring to volume.” Therefore, the mass of the sample can be calculated. In line 147. “According to the external standard method, the area of the standard solution at different concentration was detected by GC-MS.” Thus, we used 2-methyl-Anthraquinone as standard solution at different concentration because of the typical substance of teak. The relation was good with R2 of 0.9987. Thus, the quantitative value could be reliable.

There are a several typos in chemical names, and it is not clear to me how the components listed in lines 161 and 162 are isomers (chloranol (hypochloric acid), hexadecanoic acid and deoxylactam? Which deoxylactam of the many?).

Thanks. We have confirmed all the chemical names in the ChemiSpider already, and so sorry to mislead you because of our wrong statement about the documents. All of the substances with a coordinate relation, which had existed in teaks from Java Island. Please check.

Sincerely yours,

Hongyun Qiu, Ru Liu, Ling Long

May 14, 2019

Reviewer 2 Report

Comments and Suggestions for Authors:

In this paper, the acetone extractive in heartwood and sapwood of teak were analyzed by GC-MS. However, the result have some problems that should be explained.

1.      49 components and 26 components of acetone extractive were detected in heartwood and sapwood of teak by GC-MS, respectively. However, non-volatile components are difficult to detect with GC-MS, so the compounds are often derivatized. In this paper, authors detected the acetone extractive by GC-MS directly. Acetone extracts from many plant materials usually contain multiple ingredients. The authors need to explain why this extract does not contain non-volatile compounds?

2.      In figure 3, the “percentage” in sapwood of teak is less than 100%. Is there any reason?

3.      In table 5 title, “the content of acetone and petroleum ether extractive in …”, please confirm which solvent used to extract the plant material. There is only one set of data for heartwood and sapwood here.

4.      Line 186, the main extractives that ..., this sentence is not over yet, no need to break the sentence.

5.      Line 212, 9a-tetrahydro-9, 10-anthracenedione#.... What does the # symbol mean?

6.      The manuscript is not carefully written; a number of typographical and grammatical errors are present in the text.

7.      The references format must be consistent. For example, in reference 10, the author’s name is incorrectly written.

I hope that my comment is useful for the improvement of the article.

Author Response

We must thank you for the valuable comments and suggestions, which helped improve our manuscript greatly. Please do forward our heartfelt thanks to the reviewers. Based on the comments we received, careful modifications have been made to the manuscript. All changes were marked in red text. We hope that the revised manuscript answered the questions. Below you will find our point-by-point responses to the comments/ questions:

To Reviewer 2:

1. 49 components and 26 components of acetone extractive were detected in heartwood and sapwood of teak by GC-MS, respectively. However, non-volatile components are difficult to detect with GC-MS, so the compounds are often derivatized. In this paper, authors detected the acetone extractive by GC-MS directly. Acetone extracts from many plant materials usually contain multiple ingredients. The authors need to explain why this extract does not contain non-volatile compounds?

Thanks. By using GC-MS, we can first to obtained the compounds from the extractives, and predict the chromatic aberration of teak. We also took the total phenol content by spectrophotometry. But it was difficult to separate them. Other techniques, such as HPLC and UV/vis, it need to first mark the identified compounds. As you suggested, the GC-MS was sensitive for the volatile compounds, for the non-volatile compounds, we will take HPLC-MS in further study.

2. In figure 3, the “percentage” in sapwood of teak is less than 100%. Is there any reason?

Thanks. The percentage in figure 3 was the various compounds in heartwood and sapwood with their sum respectively. It means that all percentage in heartwood and sapwood were added to 100%.

3. In table 5 title, “the content of acetone and petroleum ether extractive in …”, please confirm which solvent used to extract the plant material. There is only one set of data for heartwood and sapwood here.

Thanks. We are sorry to make such mistake, the correct Table 5 title had been amended by deleting the “and petroleum ether”. Please check.

4. Line 186, the main extractives that ..., this sentence is not over yet, no need to break the sentence.

Thanks. The break had already been deleted. Please check.

5. Line 212, 9a-tetrahydro-9, 10-anthracenedione#.... What does the # symbol mean?

Thanks. It was the simple name of the substances in the results of mass spectrum, we have confirmed all the chemical names in the ChemiSpider already.

6. The manuscript is not carefully written; a number of typographical and grammatical errors are present in the text.

Thanks. We are sorry to make these mistakes, we have confirmed all the chemical names in the ChemiSpider already. Please check if they were OK now.

7. The references format must be consistent. For example, in reference 10, the author’s name is incorrectly written.

Thanks. We have carefully corrected the mistake in the reference. Please check.

Ref:

10. Zhou, Z.Z.; Liu, S.C.; Liang, K.N.; Ma, H.M.; Huang, G.H. Growth and mineral nutrient analysis of teak Tectona grandis grown on acidic soils in south China. J. Forest. Res. 2017, 28, 503-511.

Sincerely yours,

Hongyun Qiu, Ru Liu, Ling Long

May 14, 2019

Reviewer 3 Report

The authors set out to clarify the chemical color changes of teak by comparing the chemical composition of the heartwood and sapwood by GC-MS of acetone extracts of wood samples. The work is described well, with few English grammar and punctuation errors, which should be eliminated by careful editing.

However, the various standard GB/T methods of analysis should be referenced properly.

It is not clear whether the same samples were extracted successively with the different solvents, or whether a fresh samples was extracted by each solvent. Please clarify this.

Also the MS library search should be approached with caution, because mis-identification is possible and artifacts are introduced during the extraction process. For example, Phthalic acid, butyl(2-chlorocyclohexyl) methyl ester (32) is most probably a plasticizer introduced by the solvent. I also doubt whether phthalic anhydride (34) is an actual product extracted from the wood.

Please also correct the chemical names in Table 4, such as methylcyclopenlane (4), cydohexon (5), -dihudro- (17), and -benzodixol- (25).

The GC temperature program heating at 10 deg/min from 40 to 280 deg would take 24 minutes, not 15 min. Please clarify this.

Apart from these corrections, the manuscript is publishable.

Author Response

We must thank you for the valuable comments and suggestions, which helped improve our manuscript greatly. Please do forward our heartfelt thanks to the reviewers. Based on the comments we received, careful modifications have been made to the manuscript. All changes were marked in red text. We hope that the revised manuscript answered the questions. Below you will find our point-by-point responses to the comments/ questions:

To Reviewer 3:

The authors set out to clarify the chemical color changes of teak by comparing the chemical composition of the heartwood and sapwood by GC-MS of acetone extracts of wood samples. The work is described well, with few English grammar and punctuation errors, which should be eliminated by careful editing.

However, the various standard GB/T methods of analysis should be referenced properly.

Thanks. We have cited the standard in the reference [22] in our manuscript. Please check.

Ref:

22. Shi, S.; He, F. Analysis and detection of pulp and paper making, Beijing, 2009.

It is not clear whether the same samples were extracted successively with the different solvents, or whether a fresh samples was extracted by each solvent. Please clarify this.

Thanks. We are sorry I did not make this part clear, we have corrected it in line 72. “The samples were extracted by alcohol-benzene, 1% NaOH, cold water, hot water, and the acetone, respectively referring to…” Please check.

Also the MS library search should be approached with caution, because mis-identification is possible and artifacts are introduced during the extraction process. For example, Phthalic acid, butyl(2-chlorocyclohexyl) methyl ester (32) is most probably a plasticizer introduced by the solvent. I also doubt whether phthalic anhydride (34) is an actual product extracted from the wood.

Thanks. We have added the discussion for it in line 364 that “It should be noted that some compounds might not belong to wood extractives, such as estriol, a mammalian hormone, and antioxidant of butyl(2-chlorocyclohexyl) methyl phthalate. These compounds might be the dissolved impurity from the plastic lips.” Please check.

Please also correct the chemical names in Table 4, such as methylcyclopenlane (4), cydohexon (5), -dihudro- (17), and -benzodixol- (25).

Thank you very much. We have confirmed all the chemical names in the ChemiSpider already. Please check if they were OK now.

The GC temperature program heating at 10 deg/min from 40 to 280 deg would take 24 minutes, not 15 min. Please clarify this.

Thanks. I apologize for making you misunderstand our meaning, we have corrected it in line 102 that “The initial temperature was 40 °C and rose to 280 °C at 10 °C/min, then retained for 15 minutes.” Please check.

Sincerely yours,

Hongyun Qiu, Ru Liu, Ling Long

May 14, 2019

Reviewer 4 Report

Please find comments in attached.

Author Response

We must thank you for the valuable comments and suggestions, which helped improve our manuscript greatly. Please do forward our heartfelt thanks to the reviewers.

To Reviewer 4:

The manuscript reports the differences in chemical components of teak (Tectona grandis) heartwood and sapwood. From the chemical investigation, a relationship of their chemical component and color change was suggested. The experiments were well designed, and the manuscript was well-written. I found no serious flaws with the work presented. Therefore, l recommend that the manuscript should be accepted for publication in the Molecules journal.

Thank you for your affirmative reply of our manuscript.

Sincerely yours,

Hongyun Qiu, Ru Liu, Ling Long

May 14, 2019

Round 2

Reviewer 1 Report

While the authors did some improvements to the original manuscript, my initial criticism has not been overcome.

GC-MS is a very powerful method, but it is not the right choice to search for changes in wood extractives and chromophores. Wood extractives are larger molecules with some functional groups, such as tri- or tetraterpenes or triglycerides or waxes (with molar masses higher than 500 g/mol), that will not elute at 260°C from the GC. Special high-temperature methods reaching 400°C are generally used for these analyses. While several compound classes can contribute to color, anthraquinones and other quinoid systems are the main suspects for the investigated colour change  - here I agree with the authors. But analysis of these quinoids is most challenging, as their chemical properties keep changing with oxidation state, charge state and eventually polymerization. Also, their concentrations are typically by orders of magnitude lower than the LOD of a typical GC-MS system. A (normal phase) liquid chromatography method is much more adequate for this kind of analysis than GC-MS, especially with a Vis detector to detect chromophores.

Pooling the wood samples took away the option to account for any biological or environmental variation in the sample set. Biological samples can vary a lot, and therefore a lot of samples have to analysed to achieve a sound and meaningful result.

Electron Impact-MS and the available databases are incredibly useful, but they can also yield wrong results. Especially extractives are prone to that, as for example terpenes yield very similar spectra. The same is true for fatty acids or phenolics. There is now a statement about identification by Retention Indices, but no information if the found and the expected retention indices match.

Also, in EI-MS, signal intensity does not only depend on concentration but also on the detected compound. Quantification of all extracted components cannot be based on a calibration of a single molecule. The results cannot be correct.

The authors still report several compounds that are either artifacts from sample preparation or falsely identified, as they cannot impossibly be part of a tree. The most prominent example is Estriol. It is not enough to write a disclaimer stating that some compounds might not be part of wood extractives. It is the responsibility of scientists, to report true and confirmed information, and it is the responsibility of the analytical chemist to know if his result is right or maybe wrong. What should be the point of a research project, if the result is not knowledge but ambiguity? It cannot simply be left to the reader to tell true from false!

Author Response

We must thank you for the valuable comments and suggestions, which helped improve our manuscript greatly. Please do forward our heartfelt thanks to the reviewers. Based on the comments we received, careful modifications have been made to the manuscript. All changes were marked in text. We hope that the revised manuscript answered the questions. Below you will find our point-by-point responses to the comments/ questions:

To Reviewer 1:

While the authors did some improvements to the original manuscript, my initial criticism has not been overcome.

GC-MS is a very powerful method, but it is not the right choice to search for changes in wood extractives and chromophores. Wood extractives are larger molecules with some functional groups, such as tri- or tetraterpenes or triglycerides or waxes (with molar masses higher than 500 g/mol), that will not elute at 260°C from the GC. Special high-temperature methods reaching 400°C are generally used for these analyses. While several compound classes can contribute to color, anthraquinones and other quinoid systems are the main suspects for the investigated colour change - here I agree with the authors. But analysis of these quinoids is most challenging, as their chemical properties keep changing with oxidation state, charge state and eventually polymerization. Also, their concentrations are typically by orders of magnitude lower than the LOD of a typical GC-MS system. A (normal phase) liquid chromatography method is much more adequate for this kind of analysis than GC-MS, especially with a Vis detector to detect chromophores.

Pooling the wood samples took away the option to account for any biological or environmental variation in the sample set. Biological samples can vary a lot, and therefore a lot of samples have to analysed to achieve a sound and meaningful result.

Electron Impact-MS and the available databases are incredibly useful, but they can also yield wrong results. Especially extractives are prone to that, as for example terpenes yield very similar spectra. The same is true for fatty acids or phenolics. There is now a statement about identification by Retention Indices, but no information if the found and the expected retention indices match.

Also, in EI-MS, signal intensity does not only depend on concentration but also on the detected compound. Quantification of all extracted components cannot be based on a calibration of a single molecule. The results cannot be correct.

The authors still report several compounds that are either artifacts from sample preparation or falsely identified, as they cannot impossibly be part of a tree. The most prominent example is Estriol. It is not enough to write a disclaimer stating that some compounds might not be part of wood extractives. It is the responsibility of scientists, to report true and confirmed information, and it is the responsibility of the analytical chemist to know if his result is right or maybe wrong. What should be the point of a research project, if the result is not knowledge but ambiguity? It cannot simply be left to the reader to tell true from false!

Thanks. We are very appreciating for your sincere comments. Because this is a preliminary study to detect the chromatic components for teak, there were some deficiencies. We have added it in our abstract and conclusion for it. Please check.

Abstract: This paper on the issue of a preliminary study and the further work such as high performance liquid chromatography (HPLC) with UV/MS will be carried out. (Line 21)

Conclusion: This paper on the issue was a preliminary study. Since the GC-MS was not the most accurate analysis method, a liquid chromatography system like high performance liquid chromatography (HPLC) with UV/MS will be used to confirm the possible substance for the chromatic aberration in teak from China in further study. (Line 219)

Sincerely yours,

Hongyun Qiu, Ru Liu, Ling Long

May 20, 2019

Reviewer 2 Report

Comments and Suggestions for Authors:

Most of the comments have been answered, the manuscript should be accepted for publication in this form.

Author Response

We must thank you for the valuable comments and suggestions, which helped improve our manuscript greatly. Please do forward our heartfelt thanks to the reviewers. Based on the comments we received, careful modifications have been made to the manuscript. All changes were marked in red text. We hope that the revised manuscript answered the questions. Below you will find our point-by-point responses to the comments/ questions:

To Reviewer 2:

Most of the comments have been answered, the manuscript should be accepted for publication in this form.

Thank you for your affirmative reply of our manuscript.

Sincerely yours,

Hongyun Qiu, Ru Liu, Ling Long

May 20, 2019
